# Modulation of sonochemical reactions by cavitation driven thermal degradation of aqueous salts solutions

A. Troia [1] ✉, M. Gallone[2], V. Vighetto [2], F. Pellegrino[3], S. Hernández [2], V. Cauda [2] & V. Maurino[3]

Strategies for controlling or increasing the yield of radical reactions generated by ultrasonic cavitation in aqueous media have been the object of research for many years. Past studies have focused on the role of organic solvents in increasing Reactive Oxygen Species (ROS) formation or have investigated the effect of ultrasound on accelerating the OH radicals generation from Fenton reactive. More recently, piezoelectric micro-nanoparticles have shown a synergistic effect in activating specific reactions and increasing radicals production from ultrasound. Here we report the generation of ROS together with $H_2$ evolution or increase of oxidizing species during ultrasonic treatments of homogeneous concentrated aqueous solutions of simple salts as acidic phosphates, potassium sodium tartrate and alkaline nitrates. An increase in organic dye degradation efficiency, and the increase of reducing or oxidizing species compared with pure water has been found. The activation mechanism revealed a new, unexpected, approach to enhance the efficiency of sono-catalyzed reactions in aqueous media for environmental or energy applications.

Acoustic cavitation refers to the nucleation, growth, and violent implosion of microbubbles generated by high-intensity ultrasound in liquid media[1]. Under these conditions, bubbles behave as local hot spot in which extremely high temperature ( >10.000 °K) and pressure ( >10 MPa) are reached[2] leading light emission (Sonoluminescence, SL) by the formation of short-lived microplasma[3,4], which causes the breakage and ionization of gas or vapor molecules, present inside the bubbles during the collapse. In case of aqueous solutions, several radical oxidizing species (ROS) are produced, arising from the sonochemical splitting of water molecules and the formation of HO˙ and H˙ species[5,6] Acoustic streaming promote physical effects inducing high shear stress near the bubbles, with the formation of microjets, microstreaming, and shock waves, which can cause the exposure of dispersed solid particles or the injection of liquid microdroplets to these extreme conditions[7]. These physical and chemical effects form the foundation of sonochemistry, which involves the application of ultrasound to activate or accelerate chemical and physical processes[8,9]. The efficiency of a sonochemical process depends on several parameters, such as liquid nature, temperature, or in other terms, its vapor pressure, gas dissolved, ultrasonic frequency, and acoustic pressure, or in other words, ultrasound power. However, the current lack of control over this phenomenon has limited its use in energy and environmental applications due to the low efficiency of sonochemical treatments[8–11] In general, physical or mechanical effects as shear stress forces, shock waves, are dominant at low frequencies (20-80 kHz)[12,13] while at high frequencies (500kHz-1MHz) chemical effects as increase of hydroxyl radicals yields are usually reported[14]. Historically, numerous studies have aimed to increase the yield of sonochemical processes by scavenging radicals produced in the bubbles and releasing more efficient reactive species in the bulk solution. Recent models have focused on optimizing acoustic parameters (US frequency and power) to obtain efficient water splitting and hydrogen production[15]. Several investigations have reported the use of organic volatile solvents, such as alcohols or glycol, as radical scavengers in aqueous solutions, since they accumulate in the bubbles, react with OH radicals, and increase ROS release in the bulk, thereby forming longer-lived radical species[16]. However, as the solvent concentration increases, the accumulation of these molecules reduces the energy of bubble collapse, as a consequence the amount of radicals decreases and the scavenging effect becomes negligible[17,18].

Recently, several studies have revealed the possibility of increasing ROS formation or steer it toward the generation of a single product ($H_2$ or $H_2O_2$, for example) using ultrasounds[19–24]. It has been largely reported that micro or nano-semiconductors or piezoelectric particles could enhance the efficiency of ultrasonic processes for several applications in the environment, energetic, and biomedical fields[25–36]. However, the synergistic effects of piezocatalytic processes are still unclear as has also emerged in many recent works[37–42]. So far, all the studies involving mechanically driven catalysis (piezocatalysis, flexocatalysis, tribocatalysis, and sonocatalysis)[37] have

[1]Ultrasounds & Chemistry Lab, Advanced Metrology for Quality of Life, Istituto Nazionale di Ricerca Metrologica (I.N.Ri.M.), Turin, Italy. [2]Department of Applied Science and Technology, Politecnico di Torino, Turin, Italy. [3]Department of Chemistry, University of Torino, Torino, Italy. ✉e-mail: a.troia@inrim.it

mainly focused on heterogeneous aqueous systems. In this context, investigations in the homogeneous phase are lacking, probably due to the low selectivity of aqueous radical products and undesired side reactions that limit the efficiency of the process. To the best of our knowledge, studies on sono-catalyzed reactions in aqueous solutions of piezoelectric salts have not been reported. For this reason, we investigated initially the effect of ultrasound on aqueous concentrated solutions of straightforward piezoelectric salts, to examine the formation of ROS or other reactive species, that lastly have revealed the activation of unexpected scavenging mechanisms, not related to their piezoelectric properties, never reported before. We observed an increase in the selectivity on the production of reducing or oxidizing and hydroxyl radical species as a function of different salts. We found evidence that exposure of salt molecules to the high transient temperatures of collapsing bubbles allows for the activation of a new radical pathway that is able to optimize the yield and nature of sonochemical products. Here we report a series of experimental investigations that show how this mechanism may affect aqueous radical reactions both inside the bubbles, in the vapor phase, and the surrounding shell in the liquid phase.

## Results and discussions

We investigated ultrasonic irradiation at both low frequency (20 kHz) and high frequency (858 kHz) of saturated aqueous solutions of different piezoelectric salts as Potassium Sodium tartrate (PST) ($KNaC_4H_4O_6 \cdot 4H_2O$) and Dihydrogen Ammonium Phosphate ($NH_4H_2PO_4$)[43,44] and ferroelectric salts as Dihydrogen Potassium Phosphate $KH_2PO_4$ and Potassium Nitrate ($KNO_3$)[45,46] using sonochemical apparatus of Fig. S1, S2, S3 of S.I.

As quantitative measurements of OH˙ produced by ultrasound are still a matter of debate[47,48] we used spin trapping and photoluminescence probe measurements, together with gas chromatography, KI dosimetry, sonoluminescence spectra, and dyes degradation efficiency of Methylene Blue (MB), Methyl Orange (MO), and Bromophenol Blue (BB), for characterizing these solutions with respect to ultrasonic treatment in pure water.

## Increase of reducing species

In particular, low-frequency ultrasonic treatment of a concentrated (30% in weight) PST solution causes an increase in reducing species, which was initially observed by the reduction reaction of MB, rather than effective degradation (see S.I. Fig. S8 captions details*)[49]. The increase in reducing species was successively confirmed by redox potential measurements of the solution: a decrease of 30-40% with respect to the initial potential value of the solution has been observed, after 1 h of ultrasonic treatment at 20 kHz. A further confirmation has been obtained using Resazurin (see Fig. 1A), a dye probe[50], which changes its color in the presence of reducing species. However, no OH˙ has been detected using the 5,5-dimethyl-1-pyrroline-*N*-oxide (DMPO) probe in EPR measurements or using Terephthalic acid (TA) in photoluminescence investigations. To quantify the formation of reducing species, μ-GC measurements have been performed on sealed sonoreactors (Figs. S4 and S5 in S.I. and Material and method for details). It should be noted that after 15 min of sonication at 20 kHz (Fig. 1B), a remarkable increase of $H_2$ as gas reaction products with respect to sonication of pure water has been detected. Despite the low energy efficiency of this process, normalizing over time, 10 mmol/h of $H_2$ could be produced which is a

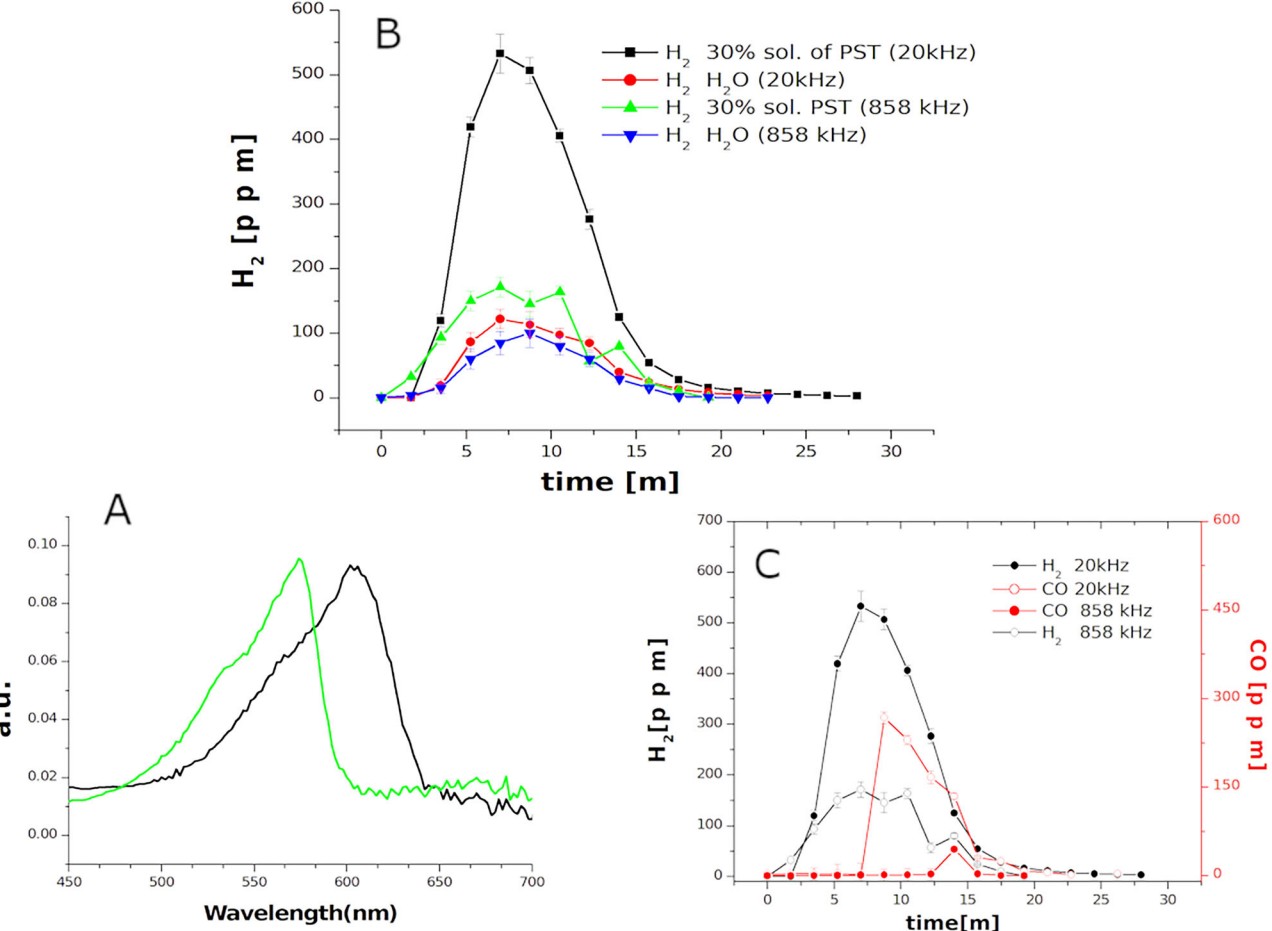

**Fig. 1 | Increase of hydrogen production from ultrasonic thermal decomposition of PST solution. A** UV-vis measurements of Resazurin reduction after sonication at 20 kHZ in a 30% PST solution. Black curve: Oxidized form of Resazurin before treatment, green curve: reduced form after treatment. **B** GC measurement of $H_2$ evolution generated from a 30% solution of PST treated at 20 kHz(black line) and 858 kHz (blue line) respect to water. **C** CO evolution, at 20 kHz and 858 kHZ (red lines) respect to hydrogen (black lines) in a 30% solution of PST.

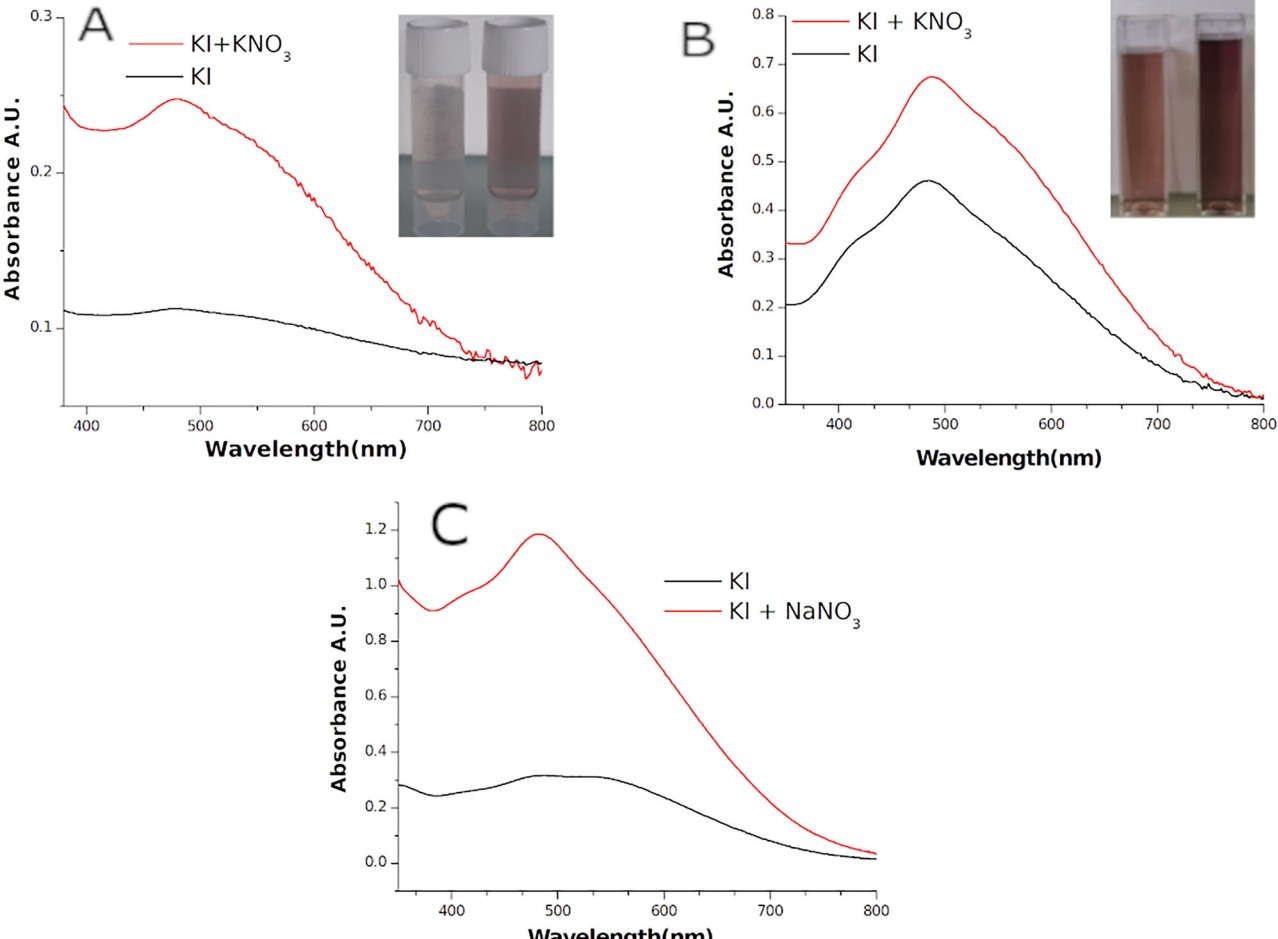

**Fig. 2 | Increase of oxidizing species from ultrasonic thermal decomposition of nitrates solutions. A** Uv-vis absorption spectra of KI 0.1 M- starch solutions treated at 20 kHz for 15 min (black line) and in presence of KNO$_3$ (20% in weight, red line). Photo in the inset shows the different color of KI solutions after US treatment with KNO$_3$ (right cuvette) or KI solution alone (left). **B** Uv-vis absorption spectra of same

solutions treated at 858 kHz for 15 min, photo in the inset shows the different color in presence of KNO$_3$ (right cuvette) or KI alone. **C** Uv-vis absorption spectra of KI 0.1 M starch solution treated at 20 kHz in presence of NaNO$_3$ (30%in weight, red line) compared to pure KI solutions.

remarkable value comparing to most of the data on hydrogen production using only ultrasound, as reported in refs. 51,52 (see table in Fig. S8 of S.I.). At 858 kHz, an increase of Hydrogen production is still visible but the amount is similar to that produced at low frequency in water. These μ-GC measurements revealed also the formation of CO (Fig. 1C, red lines) suggests a sort of tartrate dehydrogenase reductive decarboxylation of PST activated by hot collapsing bubbles[53]. As reported in thermogravimetric (TG) analysis of the thermal decomposition of PST[54] the following reaction (1), in which))) symbol represent the action of ultrasonic waves, may occur.

$$NaKC_4H_4O_6 \rightarrow ))) NaKC_3H_2O_5 + H_2 + CO \quad (1)$$

### Increase of oxidizing species

Ultrasonic irradiation of concentrated solutions of KNO3 provided further insight into this salt's activation mechanism. As no OH$^•$ has been detected from sonication of KNO$_3$ solutions both at 20 and 858 kHz using DMPO or TA probe, we have considered the well-known thermal degradation of this salt, which decompose to O$_2$ and KNO$_2$[55] and then investigated the formation of oxidizing species by different techniques: redox potential measurement, oxygen dissolved, u-GC and KI dosimetry method. As the first three methods did not allow us to establish the formation of oxidizing species we exploited the oxidation of Iodine to I$_2$[56,57] to quantify the

formation of oxidizing species in aqueous solutions, which can be easily detected with UV-vis absorbance spectroscopy of I$_3^-$ complex[47].

We exposed a 0.1 M solution of KI to ultrasound using the set-up of Fig. S1 and S3 in the S.I. Then ultrasonic treatment was repeated by adding KNO$_3$ to this solution. Since UV-vis absorption of KNO$_3$ overlap the region of I$_3^-$ complex around 350 nm (see Fig. S6 of S.I.), starch (0.1% in weight) was added to the solutions in order to evaluate the absorption of I$_2$-starch complex around 500 nm. On the Fig. 2A, B it can be seen as the solutions treated with KNO$_3$ are more violet/dark with respect to one containing KI only and UV-vis measurements confirmed this increment both at 20 and 858 kHz. Ultrasonic irradiation activates thermal decomposition of KNO$_3$, forming O$_2$ that could scavenge H$^•$ arising from water sonolysis. Then, the hydroperoxyl radical (HO2•) can react, as proposed in the following scheme (reaction 2-5) and thus increase the release of oxidizing species into the bulk, forming H2O2 reactions. Formation of NO$_3^•$ or different radical are also possible[58] but difficult to distinguish in this high ionic strength media. Finally, to confirm the thermal decomposition mechanism of nitrates we conducted similar experiment using a 30% solution of NaNO$_3$ instead of KNO$_3$ and a similar effect were observed as can be noted form the Fig. 2C.

$$(Na), KNO_3 \rightarrow ))) (Na), KNO_2 + 1/2O_2 \quad (2)$$

$$H_2O \rightarrow ))) H \cdot + OH \cdot \quad (3)$$

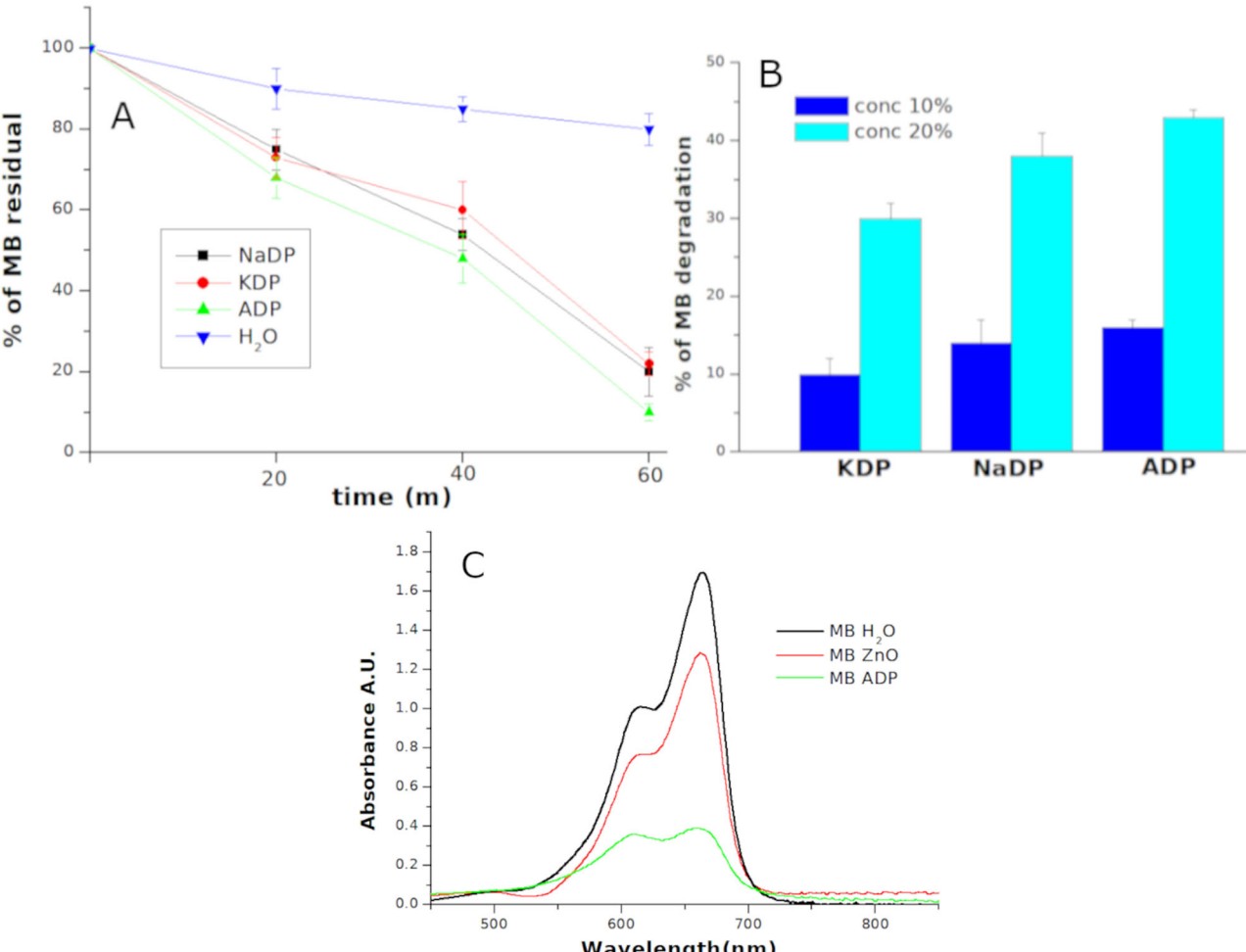

**Fig. 3 | Sonochemical degradation of Methylene Blue in acidic phosphates solutions respect to water and ZnO nanoparticles. A** Degradation of MB solutions at 20 kHz as function of sonication time in 20% aqueous solution of KDP, ADP and NaDP with respect to water. **B** MB degradation efficiency as function of different salts concentrations for an ultrasonic treatment of 20 min at 20 kHz. **C** Uv.vis spectra for MB degradation at 20 kHz in a solution containing ZnO piezoelectric nano-particles (0.2% in weight), line, compared to degradation in 20% solution of ADP solution (line). Blue curve is the MB spectrum before the treatment.

$$H \cdot + O_2 \rightarrow HO_2 \cdot \qquad (4)$$

$$H \cdot + HO_2 \cdot \rightarrow H_2O_2 \qquad (5)$$

## Increase of OH Radicals

The experiments with concentrated solutions of $NH_4H_2PO_4$ (ADP) and $KH_2PO_4$ (KDP) revealed a more complex mechanism, which involves the scavenging of aqueous $OH^\cdot$ by acidic phosphates activated by cavitation bubbles.

As the results obtained with PST and KNO3 indicate that salt's thermal decomposition would seem to be the key to these sono-catalyzed mechanisms, we included another acidic phosphate, NaH2PO4 (NaDP), which isn't piezo/ferroelectric. With these salts we observed an increase of dyes degradation efficiency for Methylene Blue (MB), Methyl Orange (MO) and Bromophenol blue (BB), with respect to the use of pure water, both at low and at high frequency. Figure 3 shows the increase of MB degradation efficiency with the 20% solutions of these salts with respect to pure water (Fig. 3A), and as a function of acidic phosphates concentration (Fig. 3B). Figure 4 shows the degradation of MO as function of different phosphates (4 A), UV-vis spectra of MO after treatment at 858 kHz in a 20% solution of ADP (Fig. 4B) and the UV-vis spectra of Bromophenol Blue after treatment

at 858 kHz in a 30% solution of ADP (Fig. 4C). In order to evaluate the efficiency of salt activated degradation a common piezoelectric material has been also tested. Piezocatalytic degradation of MB and MO with commercial ZnO nanoparticles are show in Figs. 3C and 4B compared with degradation efficiency of ADP and NaDP, respectively. After ultrasonic treatment at 20 kHz for 30 min MB degradation resulted more efficient using 30% solution of ADP than ZnO nanoparticles (0.2% in weight) Fig. 3C red curve, while at 858 kHz NaDP resulted more efficient for MO degradation after 90 min of treatment respect ZnO, Fig. 4B black curve.

However, no $OH^\cdot$ have been detected from EPR measurements, (see Fig. S7 of the S.I.). On the contrary, a variation of OH radicals has been observed with photoluminescence probe (TA) measurements. TA reacts with $OH^\cdot$ forming a TA-OH luminescing molecule and its quantification measurements have been performed as a function of different salts, their concentration, and the ultrasonic frequency. We have carried out several experiments, both at 20 kHz and 858 kHz. Examples of photoluminescence (PL) spectra of TA after sonication at 20 kHz in pure water and solutions of $KH_2PO_4$, $NH_4H_2PO_4$ and $NaH_2PO_4$ are reported in Fig. 5A. Figure 5B shows the total PL signals quantification as function of different salt concentration.

In this case, it is worth noting that the PL intensity increases with respect to pure water when using a 10% solution of KDP and NaDP, while it decreases in the presence of ADP or when a higher salt concentration is used (Fig. 5B). The experiments at 858 kHz show a similar trend, but with an

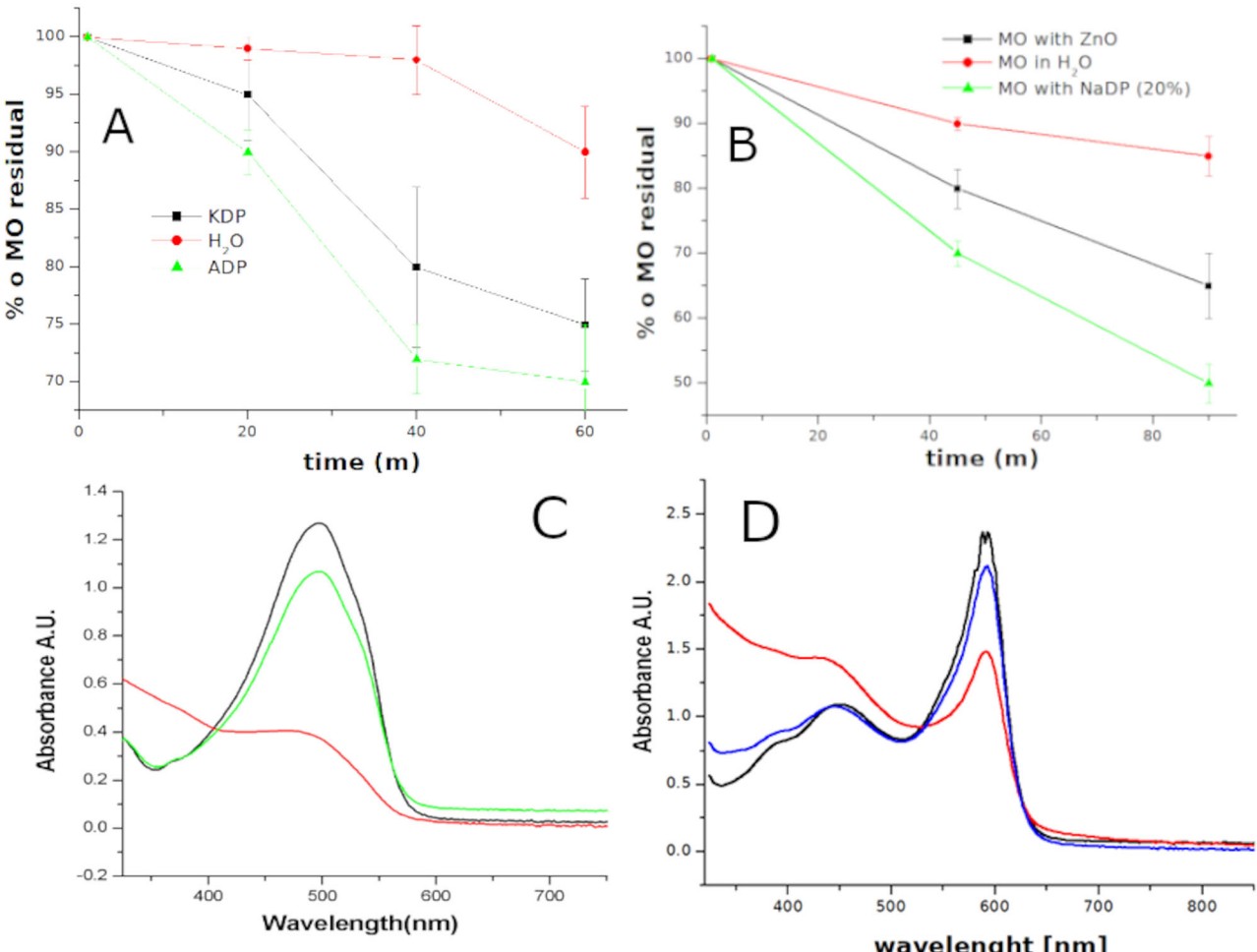

**Fig. 4 | Sonochemical degradation of Methyl Orange and Bromophenol Blue in acidic phosphates solutions. A** Degradation of MO solutions at 20 KHz as function of sonication time in 20% aqueous solution of KDP ADP and with respect to water. **B** Degradation of MO solutions at 858 kHz in presence of ZnO nanoparticles respect pure water and a 30% solution of NaDP **C** Uv-vis spectra of MO degradation in water (green line) and ADP (red line) at 858 kHz for 45 min. **D** UV-vis spectra for Bromophenol blue degradation at 858 kHz for 60 min using 20% ADP solutions (red line) respect to water (blue line).

interesting difference, as the PL intensity increases for the pure water using 10% solution of KDP, NaDP and ADP (Fig. 5C). Then the PL signal drops to the value of the water sample as the salt concentration increases, except for the ADP solution, for which a decrease is observed (Fig. 5D).

It's commonly accepted that TA underestimates the amount of OH·, since its low solubility prevents the accumulation inside the bubbles and can react only with radicals released into the bulk[59]. Since we observed an increase in dye degradation as a function of salt concentration, it appeared evident that these anomalous trends in OH· quantification, dependent on salt concentration, type of cation, and ultrasonic frequency, revealed that further radical reactions may occur.

As for the case of PST and KNO₃ we considered the thermal decomposition reactions of these salts. Acidic phosphates, such as $KH_2PO_4$, $NH_4H_2PO_4$ $NaH_2PO_4$, are used as flame extinguishers[60] since they act as scavengers of reactive radical species produced by flames, suppressing chain reactions that lead to fire propagation and explosion[61,62]. Our results indicate a similar scenario, considering collapsing bubbles as "transient hot flames" suppressed by salt molecules. Salts inhibition mechanism involves both energy absorption by salt thermal decomposition and chemical scavenging by HOPO and $HOPO_2$ species with OH· and H·. Several reaction mechanisms have been proposed[63,64] and, for more clarity, a list of possible involved reactions is reported in Table 1. As it can be seen from the Table 1, scavenging reactions by phosphates could lead to many reactive radicals[63,65–71] that could contribute to dyes degradation.

Formation of radical species from acoustic cavitation can also be detected from the emission spectra of the sonoluminescence (SL) phenomenon. Since PO· emission has been observed from sonoluminescence in $H_3PO_4$[72], we performed multibubbles sonoluminescence experiments (see SI for experimental details) in aqueous 10% solutions of KDP, ADP, and NaDP compared to SL in $H_2O$ and $H_3PO_4$ (85%) as a reference. Although the OH· and PO· radical emissions overlap in 280–350 nm region[73], qualitative results on different emission contributions in this region as a function of different salts have been carried out. In Fig. 6 the emission spectra from different liquids and solutions are shown. In case of $H_2O$ and $H_3PO_4$ the spectra confirm a series of results previously reported in the literature, such as the OH· emission in water at 310 nm (blue curve) or the CN emission at 385 nm, PO emission at 340 nm in $H_3PO_4$ (black curve)[74]. SL spectra in KDP, NaDP solutions show the respective emission from alkali atoms (around 588-590 nm for Na* and 770-772 nm for K*) while a more intense contribution in the region between 280-350 nm have been observed in ADP solution, which may indicate the formation of PO· when using this salt. This qualitative information cannot be correlated to quantitative PL measurements, as the experimental conditions were different; however, sonoluminescence spectra support the formation of different radical species in the presence of acidic phosphates.

The results reported so far demonstrate, for the first time to the best of our knowledge, that the thermal decomposition of these salts, activated by hot transient collapsing bubbles generated by ultrasound, can modify the

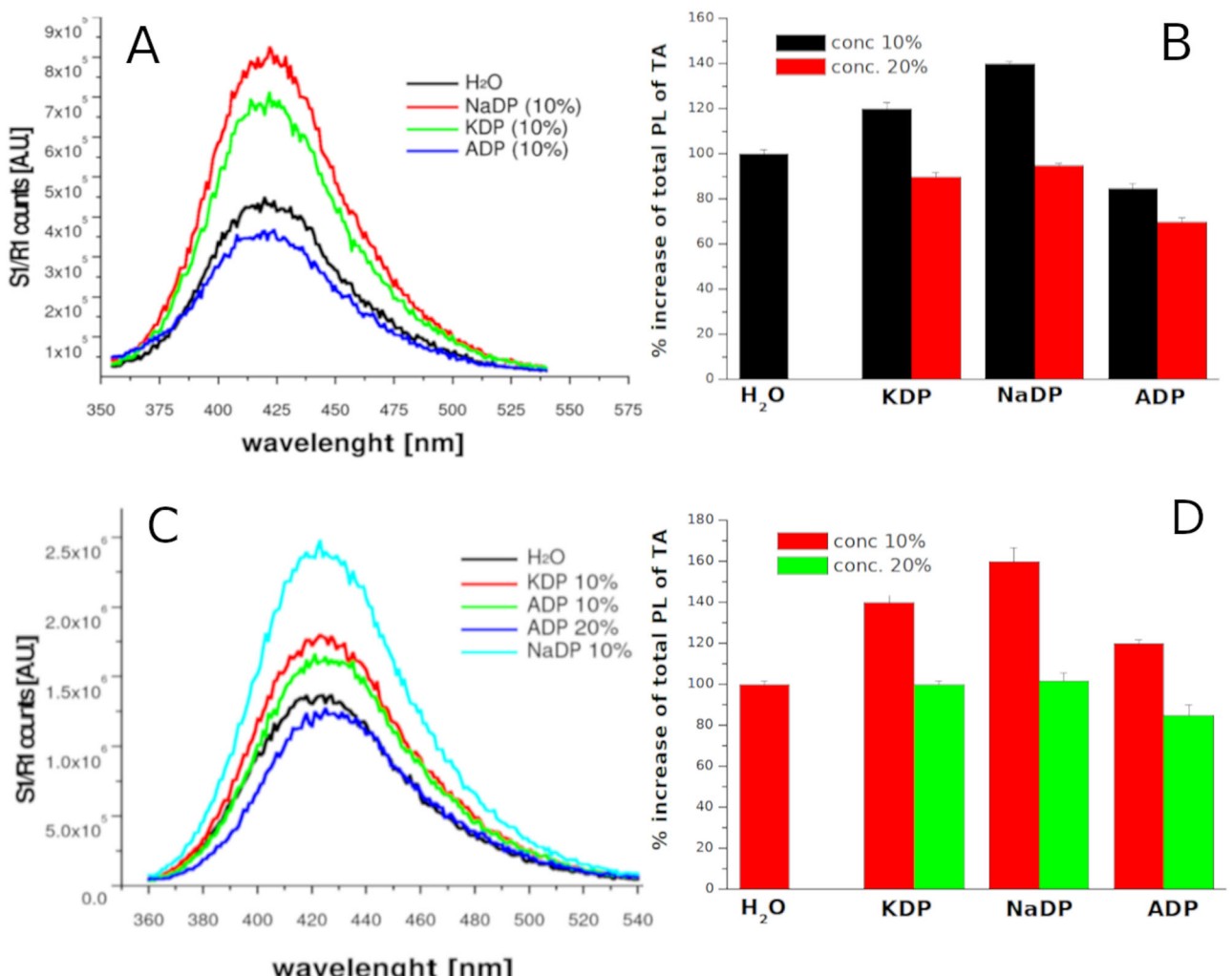

**Fig. 5 | Increase of terephthalic acid fluorescence from ROS formation in acidic phosphates solutions exposed to ultrasounds. A** Photoluminescence spectra (left) of **TA** solutions treated for 15 min with 20 KHz set-up as function of KDP, ADP and NaDP (conc 10%) with respect to water. **B** Quantified data of several spectra, as function of salts concentration. **C** Photoluminescence spectra (left) of **TA** solutions treated for 15 min with 858 kHz set-up as function of KDP, ADP and NaDP (conc 10%, also 20% for ADP) with respect to water. **D** Quantified data of several spectra, as function of salts concentration. Temperature of solutions was maintained at 25 °C.

**Table 1 | Thermal and ultrasonic ()))) reactions of water radicals and acidic phosphates radicals**

| Radical Reactions | | Application Or Study | Ref |
|---|---|---|---|
| $H_2O \rightarrow ))) \ OH^. + H^.$ | 1 | ultrasound water sonolysis | 4,5 |
| $2OH^. \rightarrow ))) \ H_2O_2$ | 2 | radical recombination reaction | 4,5 |
| $NH_4H_2PO_4 \rightarrow ))) \ NH_3 + H_3PO_4$ | 3 | Thermal degradation of ADP | 47,59 |
| $NH_3 + OH^. \rightarrow NH_2 + H_2O$ | 4 | Gas phase reaction of flame suppression for ADP | 47,59 |
| $H_3PO_4 \rightarrow ))) \ HOPO_2 + H_2O$ | 5 | Policondesation and degradation of phosphate in flame suppression | 56,57 |
| $KH_2PO_4 \rightarrow ))) \ KPO_3 + H_2O$ | 6 | Thermal degradation of KDP in flame suppression | 61,63 |
| $H_2PO_4^. + OH^. \rightarrow HPO4^{-.} + H_2O$ | 7 | Radical reaction of phosphate in self-enhanced ozonation process | 58 |
| $HPO4^. + H^+ \rightarrow H_2PO_4^.$ | 8 | Radical reaction of phosphate in self-enhanced ozonation process | 58 |
| $H_2PO_4^. + H_2O_2 \rightarrow H^+ + HO_2^. + H_2PO_4^-$ | 9 | Flash photolysis study on degradation Of Nitrobenzene by phosphate | 60,62 |
| $HOPO_2 \rightarrow PO^. + HO_2^.$ | 10 | Radical reaction of HOPO$_2$ in flame suppression | 59,66 |
| $PO^. + H^. \rightarrow HPO^.$ | 11 | Radical scavenging reaction | 63,65 |
| $HOPO_2^. + H^. \rightarrow PO_2 + H_2O$ | 12 | Radical reaction of HOPO$_2$ radical in flame suppression | 63–65 |
| $HOPO_2^. \rightarrow PO_2 + OH^.$ | 13 | Radical reaction of HOPO$_2$ radical in flame suppression | 47,66 |
| $PO_2 + H^. \rightarrow ))) \ HOPO^.$ | 14 | Scavenging reaction of H$^.$ radical in flame suppression | 64,65 |
| $HOPO^. + OH^. \rightarrow PO_2 + H_2O$ | 15 | Termination reaction of flame suppression | 57,66 |

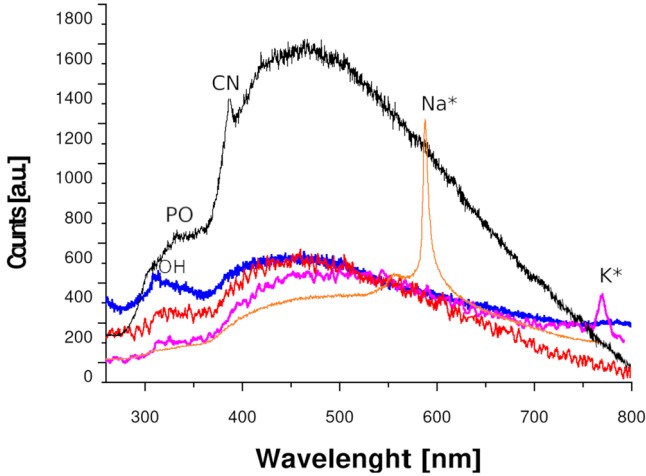

**Fig. 6 | Multibubble Sonoluminescence.** Sonoluminescence spectra of ADP (red), KDP (violet) and NaDP (orange) solutions compared to water (blue) and $H_3PO_4$ (black). The region between 300–350 nm shows the different contribution for each solution promoting $OH^\bullet$ and $PO^\bullet$ radical emission band. Emission lines form K, Na atoms and CN group are also well visible.

yield of products in sono-catalyzed reactions under homogeneous conditions. The different results as a function of frequency highlighted how this mechanism can occur both within the bubbles and at the liquid-gas interface, as recently proposed[75]. In the case in which the salt generates a reducing specie, as for PST, a higher efficiency was observed at low frequency since it is known that at 20 kHz the amount of salt injected into collapsing bubbles as small droplets is higher[76]. In the presence of $KNO_3$ an increase in oxidizing species was observed at both frequencies. Previously the increment of $H_2O_2$ from sonication of aqueous solution of $HNO_3$ $NaNO_3$, and $N_2H_5NO_3$ have been reported[77,78] but a chemical catalyzed reaction for the generation of $NO_3^\bullet$ and its reaction with $OH^\bullet$ was exploited in that case. In our experiments, the increase in oxidizing products arises from the thermal decomposition of $KNO_3$ or $NaNO_3$ by liquid droplet injection or salt exposure to a faintly warm liquid shell interface. As scavenging of $H^\bullet$ radical and hydroperoxyl radical formation are strongly favored in the liquid phase[65], this mechanism would lead to an increase in oxidizing species, since it also reduces the recombination rate of aqueous radicals.

The results obtained with acid phosphates also highlighted how this scavenging mechanism is influenced by ultrasound frequency, salt concentration and the relative cations. At low frequency (Fig. 5A, B) the increase of PL intensity in 10% solutions indicates a more efficient release of $OH^\bullet$ in the bulk by scavenging from $HOPO_2$, HOPO and relative radical mechanisms (5,6,7,9,10,11,13,14) of Table 1, probably occurring at liquid-gas interface[65]. As the salt concentration increases, a major amount of phosphate molecules enters the bubbles, lowering the release of $OH^\bullet$ in the bulk because other phosphate radicals are formed. In case of $NH_4H_2PO_4$ this mechanism is enhanced by the reactions (3,4) of Table 1 as $NH_3$ could form in the gas phase contributing to reduce $OH^\bullet$ release[61–63]. This hypothesis is also supported by the results at high frequency: as a lower amount of liquid is injected into the bubbles at 858 kHz, then the formation of $NH_3$ is much lower. As a consequence, in 10% solutions, the photoluminescence (PL) variations with respect to water for ADP follow a similar trend to KDP and NaDP, while in 20% solutions, the PL signal decreases as more NH3 could form in the gas phase. In flames suppression mechanisms, the physical energetic adsorption from thermal degradation process and from chemical radical reactions of decomposition products are involved[66,71,79]. Similarly, we could affirm that this phosphate scavenging process is affected by both physical and chemical effects of cavitation. Lastly to definitively exclude a role of the ionic strength or the high conductivity of these solutions, we performed further tests using concentrated solutions of NaCl, $K_2SO_4$ and $Na_2SO_4$ in which no meaningful enhancement of ROS

production was measured or detectable conductivity and redox potential variation. In case of PST, the decrease of redox potential is due to formation of reducing species as a consequence of its thermal degradation.

## Conclusions

In this work, we have described a series of acoustic cavitation-induced effects in homogeneous aqueous solutions of piezo/ferroelectric salts. Although a theory that correlates the chemical effects and plasma formation in cavitation to electrical discharge phenomena has been proposed in the past[80–83], and the formation of ordered structures has been observed in concentrated solutions of these salts[84–86], our unexpected results highlight the absence of a role of piezo/ferroelectric properties in these experiments.

We can state that collapsing bubbles act as transient microreactors that activate the thermal decomposition of salts molecules in concentrated solutions of tartrates, nitrates, and acidic phosphates, and in turn their decomposition products enhance the water sonolysis reactions.

Our experiments have revealed that two different scavenging regions are present: a) within collapsing bubbles and b) at liquid-gas interface. Both regions can be exploited for generating ROS, especially at high frequency, or for $H_2$ specific generation. Furthermore, this fundamental study opens plenty of room to play and tune manifold parameters, deserving future investigations, such as the optimal ultrasonic frequency to activate specific processes, the salt types, the salt concentration (i.e., supersaturated solutions), or synergistic effects with piezocatalytic micro/nanoparticles. These results represent a novel starting concept for an optimized use of ultrasonic cavitation, with possibility to control and increase the yield of sonochemical processes in homogeneous aqueous media. With potential application in energy and environmental applications.

## Materials and methods
### Ultrasonic equipment

Ultrasonic treatments at 20 kHz have been performed using a HD2200 Badelin sonotrode equipped with TT 13 tip dipped 1, 1.5 cm into the solutions, working at 45% of maximum power, corresponding to a power density 5 W/cm² as reported from our previous calorimetric measurements[87,88]. High frequency ultrasonic treatments were performed using a Meinhardt high frequency ultrasonic transducer driven by a function generator (Agilent 33250 A) and Meinhardt amplifier at 858 kHz with an acoustic pressure of 1.5 MPa measured using calibrated Müller Platte Needle hydrophone (mod 100-100-1). Double jacket water chilled reactors have been used for all the experiments which allow to maintain a temperature of 25 °C (see experimental set-up in S.I. Figs. 1 and 2). Time of ultrasonic treatments varies from 10, 20, 40,60,90 min of sonication, depending on the relative measurements to be carried out.

### Salts solutions for dyes degradation ROS analysis

Deionized MilliQ water have been used to prepare the solutions. All salts have been purchased form Merck. Salts concentration ranging from 10% up to 30% in weight depending on salt solubility. 0,025 mM solution Methylene Blue, 0,030 mM Methyl Orange and 0.033 mM of Bromo phenol Blue have been tested for dyes degradation experiments. 30 ppm solution of terephthalic acid (TA) have been used for yield measurements of OH radical through photoluminescence (PL) measurements.

### Instrumental equipment

PL has been performed using similar apparatus of Fig. S1 in SI for 20 kHZ while a modified set up (figure S3) have been used for PL measurements at high frequency. An HORIBA Gemini 180 fluorimeter equipped with double grating scanning monochromator and Xe light source working in excitation mode (310 nm as excitation wavelength) has been used and collection within the range of TA-OH emission has been performed. Several spectra have been collected for each sample using a quartz cell, and the relative uncertainty on the total area variation was less than 2%.

μ-GC measurement for gaseous products analyses have been performed using two sealed ultrasonic set-up (seeFig. S4 and S5 of S.I. for

details). Briefly, two sealed reactors (50 ml for low frequency, and 200 ml for high frequency), were connected through two 1/8 vacuum tubes to the μ-GC. Argon (Ar) has been used as a gas carrier flow, and sampling of gas production has been carried out every 5 min during continuous sonication for 20 min. The temperature of the solutions does not exceed 40 °C. Gas sampling has been performed through a Varian 490 micro-GC equipped with a Molsieve column, after purging of the cell with a continuous Ar flow for 1 h. An Argon flow rate of 25 ml¨ min´1 was used to carry the evolved gas ($H_2$ and CO in our case) from the reactor to the μ-GC.

UV-Vis measurements have been carried out by means of ThermoScientific GENESYS 50 UV using plastic and quartz cell. Each sample has been analyzed immediately after sonication using water or pure salt solutions as a blank reference. The % of degradation has been calculated by subtraction of the total area of the absorbing peak for each dye tested. Quartz and plastic cuvettes have been used depending on different measurements.

Multibubble sonoluminesce (MBSL) spectra were acquired using a set-up similar to one reported in the literature[89]. Briefly, a Titanium (3 mm of diameter) exponential probe (Bandelin MS73) was dipped into a quartz flask filled with 100 ml of different liquids saturated with Argon. MBSL was generated with pulsed ultrasound (25% of maximum power). Light was collected from the bottom with a multicore optical fibers couples to a monochromator (Acton) and analyzed with LN-cooled CCD camera (Princeton Instruments) Resolution of UV-visible spectra is 1 + /- nm, time of acquisition 120 s.

Potential redox of the solution has been measured by means of HQ40 Hach Lange multi-meter equipped with MTC301 potential measurement probe. In case of 30% PST solution the potential decrease form +180 mV to +90 mV after 45 min of sonication at 20 kHz.

The generation of OH radical was investigated using electron spin resonance (EPR) spectroscopy EMXNano X-Band spectrometer (Bruker equipped with Bruker Xenon software) associated with the spin trapping technique which itself used DMPO (5,5-dimethyl-1-pyrroline-N-oxide, Alexis Biochemicals, USA) as spin-probing agent for oxygen radicals. As an example the decrease of EPR signal at 858 kHz in presence of ADP conc 3% with respect to water is shown in Fig. S6 of S.I.

## Data availability

The data that support the findings of this study are available from the corresponding author (**a.troia@inrim.it**) upon reasonable request.

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

## Acknowledgements

The authors would like to thank Prof. R. Spagnolo for financial support of these experimental investigations and Dott. M. Pelassa for his scientific and technical contribution.

## Author contributions

A.T. conceived the study, performed the experiments and formulated hypothesis and conclusion of the work. V.C. and V.V. supported the experiments (EPR) and theoretical evaluation. S.H. and M.G support the experiments (GC) and statistical analysis V.M. and F.P. contributed to theoretical model of radical reactions and for the analysis of the measurements. All authors discussed the results and contributed to the final manuscript.

## Competing interests

The authors declare no competing interests.
