## [Transparent Peer Review file · Communications Chemistry]

Modulation of sonochemical reactions by cavitation driven thermal degradation of aqueous salts solutions

Corresponding Author: Dr Adriano Troia

Version 0:

Reviewer comments:

Reviewer #1

(Remarks to the Author)

In general, this paper will provide a very good contribution to the scientific literature exploring, as it does, some new results in sonochemistry and sonoluminescence. I have only two minor comments which would improve this paper:

The first is that the use of English should be checked because a number of minor errors need to be corrected – these occur throughout the manuscript and detract from its scientific meaning. The first examples, identified within quotation marks, are to be found in the abstract:

1. the effect of ultrasound on accelerating the OH radicals generation from “Fenton reactive”.
2. An increase in organic dye degradation efficiency, and the increase of reducing or “oxidizing species respect pure water has been found”

There are two references which, if included in the introduction, would improve the background information available to the reader:

Details could be included of the first paper to propose that metal ions could enter the cavitation bubbles and provide sodium emission as chemiluminescence

K.J. Taylor, P.D. Jarman, The Spectra of Sonoluminescence. Aust. J. Phys. 23 (1970) 319–334.

Reference to a review of the origins of extreme conditions inside a collapsing cavitation bubble which are related to nonequilibrium plasma formation inside the collapsing bubbles.

S. I. Nikitenko Plasma Formation during Acoustic Cavitation: Toward a New Paradigm for Sonochemistry, Advances in Physical Chemistry (2014) <https://doi.org/10.1155/2014/173878>

Reviewer #2

(Remarks to the Author)

The manuscript “Activation of sonocatalytic reactions in homogenous aqueous solutions of piezo/ferroelectric salts” reports the use of concentrated aqueous solutions of piezo- and ferroelectric salts to activate sonocatalytic processes, generating reactive oxygen species (ROS) and H₂ under ultrasonic irradiation. The work is potentially interesting, but several aspects require clarification or additional evidence before publication.

Major comments:

1) Mechanistic evidence: The central claim is that piezo/ferroelectric salts enable ROS and H₂ generation via a piezoelectric activation pathway. However, the data provided do not yet convincingly exclude alternative explanations such as ionic strength, solution conductivity, or cavitation enhancement due to physical properties of concentrated salt solutions. Control experiments with non-ferroelectric salts (e.g., NaCl, K₂SO₄) at comparable ionic strength are essential.

2) Quantification of hydrogen evolution: The method used for H₂ detection is only briefly described. Calibration curves, detection limits, and error bars are missing. Without these, it is difficult to evaluate the robustness of the reported yields. Please provide quantitative data with replicates and error analysis.

3) Reproducibility and experimental details: Critical parameters for reproducibility are missing: (i) Ultrasound power density and calibration method ; (ii) Reactor geometry and positioning of transducers; (iii) Temperature control during sonication. These details must be reported to allow reproduction by other laboratories. A good point is the pictures provided in SI.

4) Scope of the phenomenon: Results are shown for a limited set of salts (mostly BaTiO₃-derived). To support the claim of a general paradigm, additional examples of both piezo- and ferroelectric salts are needed, or the authors should carefully limit the claim.

(5) Environmental/energy relevance: While dye degradation is used as a model reaction, quantitative metrics (degradation rate constants, comparison with standard sonolysis, Fenton reactions, or piezo-nanoparticle systems) are missing. Without benchmarking, it is difficult to judge the actual performance and significance for environmental or energy applications.

Minor comments:

- The English requires minor editing: e.g., "respect pure water" should be corrected to "compared with pure water."
- The terms "sonocatalytic" and "sono-catalysed" should be used consistently throughout the text.
- Figures need clearer axis labels and units; error bars should be included wherever quantitative data are presented.
- The introduction should cite more recent studies on piezoelectric nanoparticles in sonocatalysis and ROS generation (2022–2024 literature is missing).
- Abstract: the phrase "a new paradigm" is too strong given the current dataset; rephrasing is advised unless additional supporting data are provided.

Conclusion:

The idea of using homogeneous piezo-/ferroelectric salts in aqueous sonocatalysis is intriguing and could open new perspectives. However, the current version lacks essential control experiments, methodological detail, and quantitative rigor. I therefore recommend major revision.

Version 1:

Reviewer comments:

Reviewer #2

(Remarks to the Author)

The revised manuscript has substantially improved compared to the original version, and the authors should be commended for the significant additional experimental work and for their overall scientific honesty. In particular, the revised conclusions now clearly acknowledge that the observed effects are not driven by piezoelectric or ferroelectric properties, but rather by thermally activated salt decomposition and radical scavenging processes induced by acoustic cavitation. This clarification considerably strengthens the scientific soundness of the study.

The work now presents an interesting and original contribution to the field of sonochemistry, especially regarding the modulation of sonochemical pathways in homogeneous aqueous solutions. However, before the manuscript can be considered for publication, several important points still need to be addressed.

Major comments:

1. Conceptual framing and scope of the manuscript (critical point):

While the mechanistic interpretation has been significantly revised and clarified, the overall framing of the manuscript remains partially inconsistent with the conclusions. The title, abstract, and parts of the introduction still strongly emphasize piezo/ferroelectric activation, whereas the revised conclusions explicitly state that these properties do not play a role in the reported phenomena.

As a consequence, the manuscript currently risks being misleading for readers expecting a study on piezocatalysis.

Recommendation:

The authors should reposition the manuscript more clearly as a study on salt-mediated modulation of sonochemical reactions via thermally activated scavenging and decomposition mechanisms, rather than on piezo/ferroelectric sonocatalysis. This may require:

- Adjusting the title,
- Further refining the abstract,
- Streamlining the introduction to reduce the emphasis on piezocatalysis and better align it with the actual mechanisms demonstrated.

This point is essential for the clarity, credibility, and long-term impact of the paper.

2. Control experiments and exclusion of alternative effects:

The addition of experiments using non-piezoelectric salts (NaCl, K₂SO₄, NaNO₃, NaH₂PO₄) is appreciated and represents a clear improvement. These results support the conclusion that the observed effects are not related to piezoelectricity.

However:

The discussion of ionic strength and conductivity effects remains largely qualitative. Quantitative comparisons between salts with similar ionic strength but different chemical reactivity would further strengthen the argument.

Recommendation:

While additional experiments may not be strictly required at this stage, the authors should clarify more explicitly in the discussion which effects are chemically specific (salt decomposition, scavenging pathways) and which are definitively excluded (conductivity, generic salting-out effects).

3. Hydrogen evolution: robustness and contextualization :

The inclusion of error bars, repeated measurements, and methodological details for H₂ quantification is a clear improvement. Nevertheless, statements such as "among the highest values ever reported" remain somewhat strong in the absence of systematic normalization (e.g., per unit acoustic power or energy input).

A brief, cautious contextualization relative to standard sonolysis (rather than an extensive benchmarking) would improve balance.

Recommendation: Please slightly temper the strongest claims or clarify the basis of comparison to avoid over-interpretation.

4. Environmental and application relevance :

The authors explain that benchmarking against Fenton or other advanced oxidation processes was beyond the scope of this work. While acceptable, the manuscript would benefit from:

- A clearer statement that the current study is fundamental/mechanistic, and
- A more cautious discussion of potential environmental or energy applications.

Overall recommendation

The manuscript now contains interesting, original, and technically solid results, and it represents a meaningful contribution to the understanding of sonochemical processes in homogeneous systems. However, a final major revision is required to:

- Fully align the conceptual framing with the revised mechanistic conclusions,
- Avoid misleading emphasis on piezo/ferroelectric effects,
- Slightly moderate the strongest performance claims.

Once these points are addressed, the manuscript should be suitable for publication in Communications Chemistry.

16th December ,2025

Authors comments to reviewers on the paper: Activation of sonocatalytic reactions in homogenous aqueous solutions of piezo/ferroelectric salts

The authors wish to thank the Referees for their accurate examination of the present work. The referee's comments, observations and suggestions stimulated re-analysis of some parts of the work, so that a few analyses were performed in order to confirm previous data and to extend the discussion with further results.

According to the Reviewers' comments, we have modified our manuscript following their recommendations. We have acknowledged the largest part of the referee's suggestions and corrections, and hereafter we motivate those which were not accepted. In these cases, we tried to increase the comprehensibility of the text with appropriate modifications.

The modifications are highlighted in green the manuscript and a detailed response to the comment is included below.

Yours Sincerely,

Adriano Troia

Reviewers' comments:

Reviewer #1 (Remarks to the Author):

In general, this paper will provide a very good contribution to the scientific literature exploring, as it does, some new results in sonochemistry and sonoluminescence. I have only two minor comments which would improve this paper:

The first is that the use of English should be checked because a number of minor errors need to be corrected – these occur throughout the manuscript and detract from its scientific meaning.

The first examples, identified within quotation marks, are to be found in the abstract:

1. the effect of ultrasound on accelerating the OH radicals generation from “Fenton reactive”.
2. An increase in organic dye degradation efficiency, and the increase of reducing or “oxidizing species respect to pure water has been found”

There are two references which, if included in the introduction, would improve the background information available to the reader:

Details could be included of the first paper to propose that metal ions could enter the cavitation bubbles and provide sodium emission as chemiluminescence

K.J. Taylor, P.D. Jarman, The Spectra of Sonoluminescence. Aust. J. Phys. 23 (1970) 319–334.

Reference to a review of the origins of extreme conditions inside a collapsing cavitation bubble which are related to nonequilibrium plasma formation inside the collapsing bubbles.

S. I. Nikitenko Plasma Formation during Acoustic Cavitation: Toward a New Paradigm for Sonochemistry, Advances in Physical Chemistry (2014) <https://doi.org/10.1155/2014/173878>

Reply: We thank the reviewer for his/her opinion on the work and the suggestions provided. We have included the suggested references and improved the clarity of our findings to provide an appropriate background for the reader and emphasize the scientific meaning. As the scavenging and “tuning” effects of aqueous sonochemical reactions have not been reported before, we are firmly convinced of the potential of these studies and the investigative prospects they would open.

Reviewer #2 (Remarks to the Author):

The manuscript "Activation of sonocatalytic reactions in homogenous aqueous solutions of piezo/ferroelectric salts" reports the use of concentrated aqueous solutions of piezo- and ferroelectric salts to activate sonocatalytic processes, generating reactive oxygen species (ROS) and H₂ under ultrasonic irradiation. The work is potentially interesting, but several aspects require clarification or additional evidence before publication.

Reply: We thank the reviewer for his/her opinion on the work and the suggestions provided. We highlighted in green all modifications added to the paper for better clarity.

Major comments:

1) Mechanistic evidence: The central claim is that piezo/ferroelectric salts enable ROS and H₂ generation via a piezoelectric activation pathway. However, the data provided do not yet convincingly exclude alternative explanations such as ionic strength, solution conductivity, or cavitation enhancement due to physical properties of concentrated salt solutions. Control experiments with non-ferroelectric salts (e.g., NaCl, K₂SO₄) at comparable ionic strength are essential.

REPLY: We thank the reviewer for this comment. As stated in the manuscript, though not clearly enough, our findings indicate that the activation mechanism follows the path of thermal degradation. We had already included some experiments with non-piezoelectric/ferroelectric salts; see the case of NaH₂PO₄ (page 8 of the original manuscript). We have excluded the effect of ionic strength or conductivity, as conductivity measurements with different probes showed no meaningful variation. However, the Reviewer's suggestion has been considered, and we have conducted additional experiments with common salts, such as NaCl and K₂SO₄. Additionally, in these cases of non-piezoelectric/ferroelectric salts, no improvement in dye degradation has been observed. In addition, we conducted further experiments on Iodine oxidation using NaNO₃ (whose degradation temperature, as well as degradation mechanism, are similar to KNO₃). The results , are shown in Fig. 2C. Even in this case ROS scavenging and sonochemical product tuning are activated by this salt, whose decomposition products contribute to the radical formation mechanism from aqueous species under ultrasound irradiation. In the revised version of the manuscript conclusions, we clearly stated these points.

2) Quantification of hydrogen evolution: The method used for H₂ detection is only briefly described. Calibration curves, detection limits, and error bars are missing. Without these, it is difficult to evaluate the robustness of the reported yields. Please provide quantitative data with replicates and error analysis.

Reply: We thank the Reviewer for this comment. More details on the used method and calibration technique have been added to the materials section. We have repeated the experiments, and error bars have been added to each graph, confirming the robustness of our results and the low error in the analysis.

3) Reproducibility and experimental details: Critical parameters for reproducibility are missing: (i) Ultrasound power density and calibration method ; (ii) Reactor geometry and positioning of transducers; (iii) Temperature control during sonication. These details must be reported to allow

reproduction by other laboratories. A good point is the pictures provided in SI.

Reply: We thank the Reviewer for the observations. We added further details of ultrasound power density and acoustic pressure measurements in materials section. We provided more detail on the reactor geometry and transducer position, including schemes of each experimental apparatus in the SI. We finally reported the temperature measurements of the sonicated solutions in our water jacket-cooled reactor, which allowed us to maintain the temperature of the sonicated solutions around 25 °C.

4) Scope of the phenomenon: Results are shown for a limited set of salts (mostly BaTiO₃-derived). To support the claim of a general paradigm, additional examples of both piezo- and ferroelectric salts are needed, or the authors should carefully limit the claim.

Reply: We thank the reviewer for this observations. We conducted additional experiments using piezoelectric ZnO nanoparticles, comparing the degradation efficiency of MB and MO with that of NH₄H₂PO₄. These experiments yielded the data shown in Figures 3C and 4B. The NH₄H₂PO₄.salts resulted in a more efficient degradation of dyes with respect to the ZnO NPs. This behavior was more evident at high-frequency ultrasound, and this would support the data reported by us in a previous paper [ref 92 “Piezo/sono-catalytic activity of ZnO micro/nanoparticles for ROS generation as a function of ultrasound frequencies and dissolved gases” Ultrasonics Sonochemistry 97 (2023) 106470] . In that work, an adverse effect due to ultrasonic absorption by solid particles reduces the synergistic effect of piezo-catalysis. In these terms, using homogeneous salt solutions would improve the efficiency of ROS formation without absorption effects.

(5) Environmental/energy relevance: While dye degradation is used as a model reaction, quantitative metrics (degradation rate constants, comparison with standard sonolysis, Fenton reactions, or piezo-nanoparticle systems) are missing. Without benchmarking, it is difficult to judge the actual performance and significance for environmental or energy applications.

Reply: We thank the reviewer for his/her comment. Evaluation of degradation constants or comparison with other methods, such as Fenton, was not considered, as the results reported in this work form the basis of a completely new, previously unreported study. However, as noted in point 4, we compared ZnO particles, widely used in piezocatalysis, with standard sonolytic processes. We can state that a significant increase in ROS production or in the production of a single species (H₂ or oxidants) in sonocatalyzed processes can be achieved using simple inorganic salts, which are low-cost and warrant further investigation.

Minor comments:

-The English requires minor editing: e.g., “respect pure water” should be corrected to “compared with pure water.”

We corrected misleading expressions throughout the paper, improving the English form.

-The terms “sonocatalytic” and “sono-catalysed” should be used consistently throughout the text.

We revised the misleading expression to be more consistent.

- Figures need clearer axis labels and units; error bars should be included wherever quantitative data are presented.

We modified the picture and graphs, adding error bars for each quantitative data presented.

- The introduction should cite more recent studies on piezoelectric nanoparticles in sonocatalysis and ROS generation (2022–2024 literature is missing).

We added further literature data of the years indicated by the referee [ref 37-42]. We found lot of studies using different piezoelectric nanoparticles but none reported the sonocatalysis effects that we observed activated by simple salts degradation.

- Abstract: The phrase “a new paradigm” is too strong given the current dataset; rephrasing is advised unless additional supporting data are provided.

We modified this expression to “new approach” because we are strongly confident that the results reported here represent a shield in an unexplored field of homogeneous sonocatalysis in aqueous media, which could be exploited in both energy and environmental applications.

Conclusion:

The idea of using homogeneous piezo-/ferroelectric salts in aqueous sonocatalysis is intriguing and could open new perspectives. However, the current version lacks essential control experiments, methodological detail, and quantitative rigor. I therefore recommend major revision.

We thank the reviewer and are confident that we have now added the necessary control experiments to demonstrate the distinctiveness of using homogeneous salt solutions.

Rebuttal letter to last referee comment:

We thank the referee for his accurate revision of the work. As suggested we modified the title in ***“Modulation of sonochemical reactions by cavitation driven thermal degradation of aqueous salts solutions”***

Referee comments, observations and suggestions have been accepted, so that a few modifications on the abstract, introduction and final claim have been modified as proposed. Specific replies are reported in red on the following document. We have acknowledged the largest part of the referee’s suggestions and corrections.

REVIEWERS' COMMENTS:

Reviewer #2 (Remarks to the Author):

The revised manuscript has substantially improved compared to the original version, and the authors should be commended for the significant additional experimental work and for their overall scientific honesty. In particular, the revised conclusions now clearly acknowledge that the observed effects are not driven by piezoelectric or ferroelectric properties, but rather by thermally activated salt decomposition and radical scavenging processes induced by acoustic cavitation. This clarification considerably strengthens the scientific soundness of the study.

The work now presents an interesting and original contribution to the field of sonochemistry, especially regarding the modulation of sonochemical pathways in homogeneous aqueous solutions. However, before the manuscript can be considered for publication, several important points still need to be addressed.

We accepted all the suggestions listed below as we are strongly convinced that as this scavenging sonochemical processes has never been reported before could represent a new theoretical approach to enhance or modulate sonochemical reactions in aqueous media

Major comments:

1. Conceptual framing and scope of the manuscript (critical point):

While the mechanistic interpretation has been significantly revised and clarified, the overall framing of the manuscript remains partially inconsistent with the conclusions. The title, abstract, and parts of the introduction still strongly emphasize piezo/ferroelectric activation, whereas the revised conclusions explicitly state that these properties do not play a role in the reported phenomena.

As a consequence, the manuscript currently risks being misleading for readers expecting a study on piezocatalysis.

Recommendation:

The authors should reposition the manuscript more clearly as a study on salt-mediated modulation of

sonochemical reactions via thermally activated scavenging and decomposition mechanisms, rather than on piezo/ferroelectric sonocatalysis. This may require:

-Adjusting the title,

-Further refining the abstract,

- Streamlining the introduction to reduce the emphasis on piezocatalysis and better align it with the actual mechanisms demonstrated.

This point is essential for the clarity, credibility, and long-term impact of the paper.

We modified the title considering also the suggestion of the editor in : **“Modulation of sonochemical reactions by cavitation driven thermal degradation of aqueous salts solutions”**

The relevance of piezoelectric/ferroelectric nature of the salt has been removed from the abstract and the introduction related to this argument has been reduced while an emphasis on the unexpected results has been added. The piezoelectric nature of the salts has been mentioned as our first selection was motivated by this criterion and only successively we considered non piezo salts which helped to find the real mechanism.

2. Control experiments and exclusion of alternative effects:

The addition of experiments using non-piezoelectric salts (NaCl, K₂SO₄, NaNO₃, NaH₂PO₄) is appreciated and represents a clear improvement. These results support the conclusion that the observed effects are not related to piezoelectricity.

However:

The discussion of ionic strength and conductivity effects remains largely qualitative. Quantitative comparisons between salts with similar ionic strength but different chemical reactivity would further strengthen the argument.

Recommendation:

While additional experiments may not be strictly required at this stage, the authors should clarify more explicitly in the discussion which effects are chemically specific (salt decomposition, scavenging pathways) and which are definitively excluded (conductivity, generic salting-out effects).

We have performed initially also conductivity and permittivity measurement of the sonicated solutions but no meaningful change has been observed. Our finding indicates that there aren't implication of ionic strength or conductivity. As we state the a difference has been observed measuring redox potential of solutions of PST because of hydrogen generation. For example similar measurements when using KNO₃ were more tricky as the increase of red-ox potential was less stable, because its dependence by of oxygen solubilization under ultrasonic treatment .We added a short description of these consideration on last part of discussion.

3. Hydrogen evolution: robustness and contextualization :

The inclusion of error bars, repeated measurements, and methodological details for H₂ quantification is a clear improvement.

Nevertheless, statements such as “among the highest values ever reported” remain somewhat strong in the absence of systematic normalization (e.g., per unit acoustic power or energy input).

A brief, cautious contextualization relative to standard sonolysis (rather than an extensive benchmarking) would improve balance.

Recommendation: Please slightly temper the strongest claims or clarify the basis of comparison to avoid over-interpretation.

We temper the claim on hydrogen production. However, considering the data of table in Figure 8 of Supplementary Information our results potentially could represent the highest, although evaluation in terms of time and power are estimated, so that further real experiments are needed to support this claim. We state our finding as remarkable considering that only ultrasound has been used.

4. Environmental and application relevance :

The authors explain that benchmarking against Fenton or other advanced oxidation processes was beyond the scope of this work. While acceptable, the manuscript would benefit from:

- A clearer statement that the current study is fundamental/mechanistic, and
- A more cautious discussion of potential environmental or energy applications.

We state more clearly in the conclusions the fundamental role of this study tempering the potential application in energy and environmental applications

Overall recommendation

The manuscript now contains interesting, original, and technically solid results, and it represents a meaningful contribution to the understanding of sonochemical processes in homogeneous systems. However, a final major revision is required to:

- Fully align the conceptual framing with the revised mechanistic conclusions,
- Avoid misleading emphasis on piezo/ferroelectric effects,
- Slightly moderate the strongest performance claims.

Once these points are addressed, the manuscript should be suitable for publication in Communications Chemistry.

We have modified the 3 points requested in this summary as indicated by referee

Sincerely,

Adriano Troia